# Phylosymbiosis in the Rhizosphere Microbiome Extends to Nitrogen Cycle Functional Potential

**DOI:** 10.3390/microorganisms9122476

**Published:** 2021-11-30

**Authors:** Mikayla Van Bel, Amanda E. Fisher, Laymon Ball, J. Travis Columbus, Renaud Berlemont

**Affiliations:** 1Department of Biological Sciences, California State University Long Beach, Long Beach, CA 90840, USA; mikayla.vanbel@gmail.com (M.V.B.); Amanda.Fisher@csulb.edu (A.E.F.); 2Biological Sciences Department, Louisiana State University, Baton Rouge, LA 70803, USA; lball4@lsu.edu; 3California Botanic Garden, Claremont Graduate University, Claremont, CA 91711, USA; jtcolumbus@calbg.org

**Keywords:** phylosymbiosis, rhizosphere, microbiome, nitrogen cycling, *Poaceae*, *Chloridoideae*, common garden experiment

## Abstract

Most plants rely on specialized root-associated microbes to obtain essential nitrogen (N), yet not much is known about the evolutionary history of the rhizosphere–plant interaction. We conducted a common garden experiment to investigate the plant root–rhizosphere microbiome association using chloridoid grasses sampled from around the world and grown from seed in a greenhouse. We sought to test whether plants that are more closely related phylogenetically have more similar root bacterial microbiomes than plants that are more distantly related. Using metagenome sequencing, we found that there is a conserved core and a variable rhizosphere bacterial microbiome across the chloridoid grasses. Additionally, phylogenetic distance among the host plant species was correlated with bacterial community composition, suggesting the plant hosts prefer specific bacterial lineages. The functional potential for N utilization across microbiomes fluctuated extensively and mirrored variation in the microbial community composition across host plants. Variation in the bacterial potential for N fixation was strongly affected by the host plants’ phylogeny, whereas variation in N recycling, nitrification, and denitrification was unaffected. This study highlights the evolutionary linkage between the N fixation traits of the microbial community and the plant host and suggests that not all functional traits are equally important for plant–microbe associations.

## 1. Introduction

Across environments, plant richness and productivity are strongly influenced by the amount of available nitrogen (N) [1,2]. Some microbes within (i.e., endosphere) and directly surrounding (i.e., rhizosphere) plant roots have evolved ways to fix the atmospheric N_2_ into bioavailable forms (e.g., NH_4_^+^) [1,3,4,5]. The so-called N-fixer guild provides bio-available N and is essential for plant growth, and thus for the functioning of the global ecosystem [5,6]. However, not all bacterial lineages can fix atmospheric N_2_. N fixation is supported by some Proteobacteria such as *Azospirillum* [7] and *Rhizobiales* (e.g., *Azorhizobium*, *Bradyrhizobium*, *Rhizobium*) [8], a few Actinobacteria (e.g., *Frankia*) [9], and some Cyanobacteria (e.g., *Nostoc*, *Calothrix*) [10,11], among others. Similarly, the denitrification process (releasing N_2_ back to the atmosphere) is supported by the denitrifier guild, including Proteobacteria lineages, such as *Micrococcus denitrificans*, *Paracoccus denitrificans*, *Thiobacillus denitrificans*, and some *Achromobacter*, *Pseudomonas*, and *Serratia*, among others [4,6,12]. In contrast, many bacterial lineages have evolved ways to assimilate and transform bioavailable N (e.g., ammonia assimilation) [4,5,13].

Although some microbial lineages contribute to specific functions in the N cycle, microbes are frequently involved in multiple processes in the same cycle (e.g., N fixation and ammonia assimilation) [14] and, directly or indirectly, in other cycles (e.g., carbon cycling) through complex interactions with other lineages in environmental microbial communities [15,16]. For example, the amount of bioavailable N released by bacteria or from N amendment has been shown to affect the rate of leaf litter decomposition (carbon cycling) by soil microorganisms [17,18]. In addition to biotic interactions, microbial communities are influenced by abiotic factors (e.g., temperature and pH) that shape their composition (i.e., taxonomic diversity) and their ability to support specific reactions and environmental services [3,17,19,20,21]. In the soil, microbes are also affected by plants, and the rhizosphere microbial community differs from the bulk soil. The precise mechanism by which plants modulate their associated microbiome is not clearly understood; however, promoting the growth of specific microbial lineages in the rhizosphere is essential for the plants to develop some immunity against deleterious microbes and to acquire essential nutrients, such as bioavailable N [3,22,23].

As some microbial guilds provide necessary services to plants and to the ecosystem (e.g., N fixation), understanding how the rhizosphere microbiome is shaped by the plant is a prerequisite to investigating the molecular mechanisms underlying this symbiosis in natural and managed systems. However, currently, most of the research has focused on understanding the mechanistic interaction between *Arabidopsis thaliana* and its root microbiome [24,25,26].

Many studies investigating the structure of microbial communities across hosts have identified significant correlations between the composition of the microbial community and the host phylogeny [3,21,23,27,28,29,30]. This so-called “phylosymbiotic” signal [31] occurs when hosts that are more closely related share more similar microbiomes. A phylosymbiotic pattern suggests that the hosts evolve characteristics (e.g., exuding nutrients, a root morphology that provides habitat, etc.) that make the hosts able to support symbiotic relationships with particular microbial lineages. A phylosymbiotic pattern in the plant root–microbiome association may mean that the host is able to control the composition of the rhizosphere community, regardless of the abiotic environment where it is growing. However, phylosymbiosis is not universally seen across host–microbiome relationships and is generally stronger in internal (e.g., gut and endosphere) than external (e.g., skin and rhizosphere) host–microbiome systems [30]. One explanation for this pattern is that hosts control the conditions in their internal systems, whereas in external systems, the host and abiotic factors interact to shape the microenvironment and the associated microbial community. While plants may impact particular microbial lineages in their rhizosphere, it is also possible that plants and their associated rhizosphere microbiomes’ composition would correlate for other reasons. For example, if plant species and microbial lineages share the same habitat, then a co-occurrence pattern would appear without specific beneficial interactions between plants and microbes [30].

Here, we investigated the phylosymbiosis of the plant–rhizosphere microbiome with specific attention to bacterial traits supporting N cycling. We tested for phylosymbiosis using chloridoid grasses (*Poaceae, Chloridoideae*) from around the world, as grasses are known to efficiently take up N [32]. Specifically, we grew 26 species of chloridoid grasses, plus one danthonioid (*Danthoniodeae*) outgroup species, in a common garden experiment at the California Botanic Garden to test if variation in the rhizosphere’s microbial community composition (taxonomic diversity) and functional potential for nitrogen cycling correlated with the phylogenetic distance among plants.

As the plant’s ability to select and control microbes in the rhizosphere is a complex process, likely involving multiple genes [22,26,33,34], we predicted that closely related plants, with a shared evolutionary history and thus functional traits, would display similar abilities and preferences for selecting and promoting microbial lineages in the rhizosphere. In consequence, closely related plants from the same species or genus would display similar microbial communities, whereas more distantly related plants would have more distinct microbial communities. As all the plants were grown and maintained in the same conditions (i.e., a common garden), the observed differences in microbial community composition would primarily indicate the plant’s ability to modulate its rhizosphere microbiome. Alternatively, if we found that the distribution of taxa in the rhizosphere microbiome did not correlate with the plant phylogeny, this could be explained by the random drift of microbial communities between pots or other unmeasured factors affecting the rhizosphere community. In this case, the microbiome of closely related plants (e.g., the same plant species or genus) would be random or less similar than the microbiome of distantly related plants.

We next investigated the rhizosphere microbiome of the 27 grasses for variation in functional potential across microbiomes, with particular attention to annotated traits supporting N cycling. We hypothesized that fluctuation in overall N cycling potential would correlate with variation in the rhizosphere’s microbial community composition and with plant phylogeny. Indeed, as most functional traits are not randomly distributed among microbial lineages, changes in the microbial community composition often reflect changes in the functional potential [4,20]. Specifically, we investigated the distribution of 44 individual traits in eight biochemical pathways supporting N cycling in the rhizosphere. We hypothesized that variation in the traits and pathways across microbiomes would reflect their distribution in microbial lineages and their relevance to ecosystem functioning. Specifically, the distribution of phylogenetically constrained traits (e.g., N fixation) across microbiomes was predicted to mirror variation in the microbial community composition and potentially the plant phylogeny, whereas the distribution of abundant and broadly distributed traits (e.g., ammonia assimilation) was predicted to remain stable across plants and microbiomes.

## 2. Materials and Methods

### 2.1. Plant Collection and Phylogeny

In total, 26 chloridoid grass species and one outgroup (*Danthoniopsis ramosa*, *Danthonioideae*) were collected as seeds from various locations around the world (Figure 1A, Appendix A). All plants were propagated in 3.8- or 7.6-L (1 or 2 gallon) pots with a 1:1:1 ratio of unsterilized perlite:sand:finely screened peat in a greenhouse with ambient light at the California Botanic Garden (Claremont, CA, USA) (Figure 1B). The grasses’ phylogeny was inferred on the basis of the sequences of three chloroplast loci (*ndhF* and the *trnC*-*rpoB* and *trnL*-*trnF* intergenic spacers, missing *ndhF* for *Muhlenbergia brevigluma* and *Triraphis andropogonoides*). Nucleotide sequences were retrieved from publicly available datasets or sequenced for this study (Appendix A). RAxML 8.2.11 [35] was used to estimate the maximum likelihood (ML) phylogeny using the GTR + I + G model [36] rooted with the danthonioid outgroup, and branch support was assessed with a maximum likelihood bootstrap analysis (1000 replicates).

### 2.2. Rhizosphere Metagenome Sequencing

Replicated samples (*n* = 2) of rhizosphere soil (1.5–2 g) were collected by harvesting the soil loosely adhering to the root of each plant (a total of 27 plants and 54 samples). After removing the plant material and larger debris, total DNA was extracted using the DNAeasy PowerSoil Kit (Qiagen, Hilden, Germany), sheared using a focused Covaris M220 Focused-ultrasonicator (Covaris, Woburn, MA, USA), tagged using the TruSeq Nano DNA Kit (Illumina, San Diego, CA, USA), quality checked on the Agilent 2100 Bioanalyzer (Agilent Technologies, Palo Alto, CA, USA), and sequenced on an Illumina HiSeq 2500 (PE100) at the UCI Genomics High-Throughput Facility (University of California Irvine, Irvine, CA, USA). DNA sequences were uploaded to MG-RAST for repository and processing [37]. Sequences for the 54 metagenomes are publicly available on MG-RAST (mgp82531, Appendix A).

### 2.3. Nitrogen Cycle Protein Database

In order to identify the functional potential related to N cycling, we selected a subset of 44 orthologous protein families (Appendix A), as previously described [4]. First, the ID for each protein family was retrieved from KEGG using the KOFam number [38]. Next, the protein IDs for each family were obtained from a LinkDB search and the corresponding protein sequences were downloaded from UniProt [39]. The metadata for each protein sequence was amended with the corresponding functional information from KEGG to distinguish the sequences associated with each specific pathway [38,40]. The resulting sequence database, hereafter called the N Cycle Protein Database (NCPD), contained the complete protein sequences, metadata, and protein family ID for 39,743 proteins involved in N cycling (Appendix A).

For sequence length normalization, the consensus sequence length for each KO family of interest (Appendix A) was derived from the KOFam file retrieved from the KEGG database and from the multiple sequence alignment used to define the KO protein families [38]. We used the “length of the consensus positions” value as a proxy for the sequence length. Next, the count for each protein family was multiplied by the ratio of the specific protein length by the length of the longest protein family identified (i.e., glutamate synthase K00264, 2121 amino acids).

### 2.4. Metagenome Annotation

The taxonomic and functional annotation of the sequenced metagenomes was based on the MG-RAST annotation pipeline. The distribution of the taxonomic hits (from phylum to genus) was retrieved for each of the 54 datasets and used for investigating the microbial community composition. For the functional annotation of the N cycling traits, we adapted the MetaGeneHunt approach [41] to obtain a gene-level functional prediction. Briefly, we first identified all the N cycling traits in the M5nr reference database by BLASTing all the sequences from the NCPD database (an E value of ≤10^−5^ and sequence coverage >70%); the M5nr database is a non-redundant database used by MG-RAST for taxonomic and functional annotation [42]. While identifying the sequences matching with known sequences for N cycling in M5nr, we created a custom “reference annotation table” for nitrogen cycling traits (RAT-N). This non-redundant subset of the M5nr database contained 723,005 unique proteins, each with a unique MD5id. Next, sequences matching N cycling traits were identified in each metagenome by searching RAT-N entries in the MG-RAST annotation file system (i.e., file #.650) [43]. For each sample, the search result contained the protein MD5id (from the M5nr database), the matching protein family ID from the NCPD, and the hit counts in the sequenced microbiome. This information provided the direct sequence counts for traits involved in N cycling in each sequenced microbiome. For normalization and rarefaction, we used the total number of sequences (post QC) from each dataset (Appendix A).

### 2.5. Data Analysis and Statistics

Data were processed in the R software environment (version 3.6.3) using the “reshape2” (V1.4.3), “dplyr” (V0.8.3), and “vegan” (V2.5-6) packages, whereas visualization was carried out using “gplots” (V.3.0.1.1) and “ggplot2” (V3.2.1). Specifically, microbial community composition and functional potential were rarefied and analyzed using Shannon diversity and Bray–Curtis dissimilarity indexes, as implemented in the vegan package.

In all of the analyses, except when otherwise stated, one of each of the replicated samples was excluded to avoid a confounding effect of the pots on the plants’ phylogenetic signal. Correlations (Pearson and Spearman) and linear regression were computed using R base and Hmisc (V4.4.0). The effect of the sample’s geographic origin on the microbial community structure and functional potential was investigated by comparing the pairwise Bray–Curtis dissimilarity for plants derived from the same vs. distinct continents. The comparison was analyzed using Welch’s *t*-test.

## 3. Results

### 3.1. Plant Phylogeny

Our sampling of 26 chloridoid species represents the phylogenetic breadth and much of the geographic extent of the subfamily (Figure 1A, Appendix A). The maximum likelihood phylogeny (Figure 1B) resolved the Centropodieae (*Centropodia mossamadensis*) and Triraphideae (*Triraphis andropogonoides*) as the earliest-diverging lineages, and Eragrostideae as sister to a clade of Zoysieae and Cynodonteae. Relationships within Cynodonteae had the least support. The taxa represented by multiple samples, *Distichlis* (Argentina, USA), *Sporobolus* (Argentina, South Africa), and *Leptochloa dubia* (Argentina, USA), were each confirmed as monophyletic.

### 3.2. Microbiome Composition

Change in rhizosphere microbiome composition, after rarefaction, correlated with the plant phylogeny. First, as expected, within pots, microbiomes displayed more similar taxonomic composition (at the genus level) than across pots (Welch’s *t*-test, *p* < 0.001) (Figure 2A and Appendix A). Comparisons between samples from the same pot were removed from all subsequent analysis. Next, we focused on the microbiomes of plants from the same species (i.e., two collections of *Leptochloa dubia*) and the same genus (i.e., two *Distichlis* species and three *Sporobolus* species) We found that microbiomes from different pots but from closely related plants had more similar microbiomes than pairs of randomly selected microbiomes. Specifically, pairwise comparisons between samples from different pots revealed that variation in the microbiome composition was in the range of variation observed between same-pot replicates, except for one *Sporobolus* sample (Figure 2A). However, when we considered all the pairwise comparisons, after excluding the comparison among replicates, the dissimilarity between pairs of rhizosphere microbiomes increased with the phylogenetic distance among the plant hosts (r_Pearson_ = 0.2, *p* < 0.001; r_Spearman_ = 0.22, *p* < 0.001, Figure 2A). This overall weak correlation suggests that a large fraction of the microbiome is either conserved across samples, as expected in a common garden experiment, or does not mirror the plant phylogeny. Finally, we used a linear regression (MicroComp ~PlantPhylo) to assess the linkage between the variation in the microbial community composition and the phylogenetic distance among pairs of plants (slope = 0.658, *p* < 0.001, std.error = 0.063; Figure 2A). Together, these results show that more closely related plants have more similar rhizosphere microbial communities. However, the microbial communities from replicated soil samples from the same pot and samples from the same species/genus, although highly similar, were not identical (Figure 2A).

We also investigated the effect of sample geography based on where the plant was originally collected but found no significant difference when investigating pairwise comparisons of the microbiome from plants derived from the same continent and plants from different continents (Appendix A).

Across samples, a total of 1,276,324,785 sequences (post QC) were analyzed (Appendix A) and the rhizosphere microbiome was dominated by Proteobacteria, with annotated sequences ranging from 51.2 to 69.6%. This was also reflected at the genus level, as the five most abundant genera, *Bradyrhizobium*, *Rhodopseudomonas*, *Burkholderia*, *Mesorhizobium*, and *Opitutus* (phylum Verrucomicrobia), accounted for an average of 15.56% of the annotated reads (Appendix A). After rarefaction of the taxonomically identified sequences (*n* = 3,127,337 sequences/sample), 481 bacterial genera were identified. Although some abundant lineages were identified, most genera were associated with a low number of sequences. Globally, the Shannon diversity index (*H’*), computed at the genus level, ranged from 4.95 to 5.40 (Appendix A). Across samples, most genera accounting for >0.5% of the reads displayed low variation in their frequency (Appendix A). For example, among the Actinobacteria, *Frankia* accounted for 0.7% of the reads and was the most consistently detected genus across samples (coefficient of variation, CoV = 0.23), whereas the more abundant *Mycobacterium* and *Streptomyces*, accounting for 2.2 and 1.7% of the reads, had a variation of 0.56 and 0.34, respectively (Appendix A). In the Proteobacteria, among the major groups identified, many Rhizobiales were found to be consistently abundant across samples and displayed a low CoV (Appendix A). Conversely, the less abundant members of the Sphingomonadales, Burkholderiales, and Xhantomonadales orders displayed high variability across samples (Appendix A). This highlighted that the overall variation in the microbiome composition reflected changes in the frequency of the abundant bacterial lineages forming a core microbiome and the change in the taxonomic composition of the less abundant lineages.

Next, sample replicates were combined and analyzed using NMDS to investigate the clustering of the plant–rhizosphere microbial communities and the distribution of bacterial genera across plants (Figure 3 and Appendix A). Most Actinobacteria formed a distinct and homogeneous cluster overlapping only with *Ktedonobacter* (phylum Chloroflexi) and a few Proteobacteria (e.g., *Sphingobium*, *Sphingomonas*). Most of the Proteobacteria, including delta/epsilon-Proteobacteria (e.g., *Stigmatella*, *Myxococcus*), formed a large cluster overlapping with the Bacteroidetes (e.g., *Flavobacterium*, *Pedobacter*). Members of the Verrucomicrobia phylum, including the abundant *Opitutus*, clustered with the Proteobacteria. Most genera from the other phyla, including Firmicutes (e.g., *Lactococcus*, *Thermosediminibacter*), Cyanobacteria (e.g., *Nodularia*), and Chloroflexi (e.g., *Oscillochloris*), produced a large cluster (Figure 3 and Appendix A) differentiating them from the other aforementioned clusters along the first axis. The Actinobacteria cluster segregated from the Proteobacteria/Verrucomicrobia cluster along the second axis.

### 3.3. Functional Potential for Nitrogen Cycling

Across the sequenced microbiomes, variation in the microbial community composition (at the genus level) and the overall distribution of sequences supporting N cycling were highly correlated (r_Pearson_ = 0.72, *p* < 0.001; r_Spearman_ = 0.83, *p* < 0.001), thus confirming that a change in microbial community composition affects the functional potential for N cycling (Figure 2B). The plant origin had no significant effect on the relationship between the microbial community composition and the functional potential for N cycling (Appendix A).

Sequences for N cycling pathways accounted for 3.4 to 3.9% of the annotated reads. After rarefaction (*n* = 1,207,202), identified sequences involved in N cycling were not evenly distributed (Appendix A). The most prevalent sequences were for the ammonia assimilation and assimilatory nitrite to ammonia pathways. Specifically, short-read sequences matching long glutamine synthase (K01915, 529 aa), glutamate synthase (K00266, 610 aa), asparagine synthase (K01953, 529 aa), and glutamate dehydrogenase (K00261, 534 aa) together accounted for ~43% of the identified sequences for N cycling. Conversely, sequences for N fixation and denitrification accounted for ~3.25% and 0.43% of the identified sequences, respectively. As the frequency of target sequences in the short-read metagenome reflected both the abundance of the sequence of interest in the sample and their length, we normalized the sequence count according to the corresponding domain length in the KEGG reference database. Accounting for the size of the targeted sequences revealed that the frequency of short sequences for the nitrite reductase small subunit (K00363, 175 aa), nrfC (K04014, 334 aa) and nitrogenase nifH (K02588, 293 aa) tended to be underestimated relative to the longer sequences, including the nitrite reductase large subunit (K00362, 1096 aa), among others (Appendix A).

Within samples, sequences for the same pathway generally displayed a consistent distribution (Figure 4). For example, the frequency of sequences for assimilatory nitrite to ammonia correlated with each other. This pattern was also identified for sequences in less abundant pathways such as the dissimilatory nitrate to nitrite and the denitrification pathways.

We next investigated sample clustering based on the distribution of the sequences for N cycling pathways (Figure 4). The functional clustering produced two major clusters. In Cluster I, we found traits for assimilatory nitrate to nitrite, assimilatory nitrate to ammonia and dissimilatory nitrate to nitrite were overabundant relative to Cluster II. Conversely, Cluster II was enriched in traits for ammonia assimilation. Finally, the distribution of traits supporting the denitrification, dissimilatory nitrate to ammonia, and nitrification pathways were more evenly distributed between the two clusters. Replicates tended to cluster together (*p* < 0.001). Combining the replicate samples and performing the NMDS analysis (Appendix A) revealed that most pathways were conserved across plants, with the exception of N fixation and dissimilatory nitrate to ammonia.

Across samples, sequences for the nitrification pathway had consistently high variability (CoV > 0.5), whereas in some pathways, only a few sequences were variable, such as anfG nitrogenase (K00531, N fixation) and nrfA-nitrite reductase (K03385, dissimilarity nitrite to ammonia) (Appendix A). Besides these sequences, the frequency of most identified functions displayed low variability across samples (CoV < 0.25). For example, sequences for nifH nitrogenase (K02588) were extremely conserved across samples (CoV = 0.07) (Appendix A). This suggested that the functional potential to support the various N cycling pathways, although not evenly distributed, is not random.

### 3.4. Phylosymbiosis and N Cycling Traits

Considering all the traits together, the variation in the distribution of sequences supporting N cycling is weakly correlated with the phylogenetic distance between pairs of plants (R_Pearson_ = 0.04, *p* = 0.049, R_Spearman_ = 0.09, *p* < 0.001) (Figure 5A,B). However, as not all the pathways were evenly distributed across plants (Appendix A), we next investigated the correlations and linear regressions between plant relatedness and the variation in the distribution of each individual pathway (Figure 5, Table 1). When considering individual pathways, the variation in sequence distribution correlated with plant phylogenetic distance, with the exception of the traits for denitrification, which were evenly distributed across samples (p_Spearman_ = 0.18), and the two traits for the assimilatory nitrate to nitrite pathway (p_Spearman_ = 0.12) (Figure 5A).

Across samples, variation in the frequency of sequences for each individual pathway was strongly correlated with changes in the microbial community composition, except traits for the nitrification pathways. In addition, variation in the distribution of sequences for most pathways correlated with each other, except sequences for assimilatory nitrate to nitrite (Figure 5A).

The distribution of sequences for N fixation was the pathway most affected by the plant phylogeny and one of the pathways least affected by the microbial community composition (Table 1, Figure 5C,D). On the other hand, the distribution of the abundant sequences for the ammonia assimilation pathway were only marginally affected by both the plant phylogeny and the microbial community composition (Table 1, Figure 5E,F).

## 4. Discussion

The plant’s ability to control the root-associated microbial community is well established [44,45]. However, we have not fully explored how plant evolutionary variation plays a role in the rhizosphere community composition. Here, we used a common garden experiment to link plant relatedness to their ability to control the root-associated microbial communities. To our knowledge, this is the first time that this approach has been used at this scale with plants sampled from around the world.

We used chloridoid grasses grown in a common garden experiment to reduce environmental heterogeneity and to investigate how the phylogeny of these cosmopolitan grasses affects the structure and function of the rhizosphere’s bacterial microbiome. We allowed the microbial community to develop from the greenhouse environment, the soil mixture, and the seeds. Although all the grasses were maintained in unnatural greenhouse conditions and received the same treatment (e.g., water, soil), none displayed abnormal or delayed growth. The subfamily Chloridoideae contains approximately 1700 species known to be adapted to mostly arid habitats [46]. This lineage of grasses seems to have originated in Asia or Africa and has its center of diversity in Africa. Chloridoids have since spread to the other continents multiple times, with radiations in North America, South America, and Australia [46]. Although some of the taxa had been characterized before, many were sequenced for the purpose of this study. The resulting phylogeny of the 26 chloridoid grasses and the outgroup, *Danthoniopsis ramosa* (Figure 1A), corroborated the relationships found in previous studies that sampled more taxa and loci [46,47,48]. Importantly, species sampled in duplicate and species in the same genus were monophyletic.

The rhizosphere microbiome was investigated using metagenome sequencing. Replicate samples (from the same plant) and samples from closely related plants (same species or genus) displayed a highly similar bacterial microbial community composition (Figure 2, Table 1). This validated our experimental design and suggested that the rhizosphere’s microbiome assembly is not purely random. Moreover, dissimilarity in the taxonomic composition of the rhizosphere’s bacterial microbiome was correlated with the plants’ phylogenetic distance (Figure 2, Table 1), thus suggesting a phylosymbiosis between chloridoid grasses and their rhizospheres’ microbiomes. Thus, the grass species have inherited the tendency to associate with particular bacterial lineages. In the studied rhizosphere, some bacterial lineages remained abundant and formed a “core microbiome” (accounting for ~30% of the reads; Figure 3). This core microbiome could reflect a signal conserved across all the studied plants, an external constraint imposed by our experimental design (e.g., soil type), or a combination of both. However, besides the core microbiome, each plant had a unique bacterial community with many rare microbial lineages and whose precise variation (composition and abundance) was correlated with the plant phylogeny. This confirmed that, even in unnatural conditions, chloridoid grasses control the composition of their root-associated microbiome.

Understanding the precise mechanism that plants use to select their rhizosphere remains a major challenge [22]. Some studies have investigated the microbial community composition using amplicon sequencing and inbred plants to explain the variation in plant phenotype (e.g., resistance to desiccation) [3,26,34]. Interestingly, in 2018, Fitzpatrick et al. analyzed the composition of the rhizosphere and endosphere microbiome in 30 distantly related angiosperms and identified no correspondence between the rhizosphere and the plant phylogeny. Only the endosphere’s microbiome composition was strongly affected by the plant phylogeny, although the endosphere’s and the rhizosphere’s microbiome compositions were correlated [3]. These previous approaches highlighted the importance of the microbial community for plant health and revealed plants’ genetic determinants supporting the microbial community assembly. However, as genetic material and functional potential vary greatly among microbial lineages, it is essential to investigate how these structural variations relate to change in functions. Although this can be achieved through the targeted amplification of specific genes (e.g., *nifH*) [49,50], whole-metagenome sequencing provides a much broader picture, allows for the combined analysis of the composition and the functional potential, and allows us to compare multiple traits.

We found that changes in the microbiome composition mirrored the variation in the overall distribution of functional traits supporting N cycling (Figure 4). The functional potential for N utilization and N cycling is not randomly distributed in microbial communities, just as there is a non-random pattern in carbon utilization across ecosystems more broadly [20]. This is further supported by the clumped distribution of traits for N fixation in the sequenced bacterial lineages [51], and suggests that the overall distribution of the other traits for N cycling are, to some extent, also phylogenetically constrained in microbes. Since analyses of the relationship between overall microbiome composition and overall functional potential are biased towards the most abundant traits (i.e., ammonia assimilation), we analyzed each N cycling pathway independently. Pathways producing bioavailable N (e.g., N fixation and sequential conversion of nitrate to nitrite and ammonia), processes contributing to the recycling of N (e.g., ammonia assimilation), and processes causing the loss of bioavailable N (e.g., denitrification) have different impacts on the microbiome, on the plant, and thus on the whole ecosystem’s functioning. The distribution of all the traits, except processes supporting the production of nitrite, showed some degree of microbial phylogenetic constraint, as the variation in each functional potential correlated with the microbial community composition. However, only a few N pathways were also affected by the plant phylogeny (Figure 5). Not all the investigated N processes are under selection or resulted in a phylosymbiotic pattern.

Sequences for pathways supporting the production of bioavailable N are essential for the whole ecosystem yet are among the least abundant sequences in chloridoid grass rhizospheres. This is consistent with previous findings [4,5]. This also suggests that the abundance and distribution of N-fixing bacteria are restricted. Among other reasons, the amount of unavailable N and the availability of other nutrients (e.g., phosphorus) are essential factors affecting the development of N-fixers [52,53]. In addition, predatory invertebrates [54] and microbial antibiotic production [55,56] also have adverse effects on N-fixers.

At the molecular level, N fixation is supported by highly conserved molybdenum-nif nitrogenase multiprotein complexes encoded in the *nif* operon [52,57]. In addition, two alternative nitrogenase systems are known, namely vanadium-vnf nitrogenase and the alternative-anf nitrogenase. These nitrogenases share similar proteins; however, the two alternative complexes contain an additional protein (i.e., vnfG and anfG). Nitrogenase complexes are not consistently distributed across microbial lineages, with most N-fixers having one or two of these operons [58]. The alternative nitrogenases likely evolved in response to environmental variations in metal availability [59] and are activated under Mo limitation [60]. The nif nitrogenases are the best described and their sequences are abundant in reference databases. Indeed, as of July 2020, the KEGG gene database identified >800 nif, ~50 anf, and only 30 vnf nitrogenases [38,40]. We thus focused on nif nitrogenases but also identified sequences for anfG (K00531, 117 aa) as a proxy to investigate the presence of alternative nitrogenases. However, as only a few sequences for anfG were detected in some samples, their distribution will not be further discussed here. Regarding the relative abundance of nif nitrogenase, the sequences for nifH (K02588, 293 aa), nifK (K02591, 458 aa), *nifD* (K02586, 477 aa), and *nifW* (K02595, 124 aa) are unevenly distributed within and across samples. This reflects the complex evolution of *nif* nitrogenase. Indeed, among the others, *nifH* has a distinct evolutionary history relative to the other *nif* genes and is not always located within the nif operon [61]. In addition, several genomes contain multiple copies of *nifH* (e.g., *Rhodopseudomonas palustris*) [58,61,62,63]. Finally, the potential for N fixation in sequenced bacterial genomes has shallow phylogenetic conservatism (an average clade depth of ~0.018 16S rRNA distance) [51]. Together, these points suggest that N-fixers have discrete signature sets of sequences supporting N fixation. In consequence, although one could expect to find a consistent and stable potential for N fixation, the variation in the distribution of *nif* sequences identified here suggests that plants select for N-fixing lineages endowed with specific sets of genes supporting N fixation.

The distribution of the more abundant traits for the conversion of nitrate to nitrite and nitrite to ammonia across samples mirrors their abundance in sequenced genomes, with KEGG genes identifying >3000 *nirB* nitrite reductase (K00362) and >300 nitrate reductase (K10534), among others [38,40]. Despite being abundant and potentially beneficial to both the rhizosphere’s microbial community and thus to the plant host, the distribution of these sequences does not correlate with the plant phylogeny (Figure 5, Table 1).

Next, the distribution of sequences for ammonia assimilation was less variable across samples than the sequences for less abundant pathways (Figure 5, Table 1). However, even within the ammonia assimilation pathway, sequence abundance was variable. This suggests some variation in gene content in the microbial lineages inhabiting the rhizosphere and highlights how essential the N recycling function is for the microbiome and the plant. Globally, the functional potential for ammonia assimilation, although correlated with the microbial community composition, is only marginally affected by the plant phylogeny (Figure 5, Table 1).

Next, the distribution of traits for nitrification and denitrification fluctuated as the microbial taxonomic community changed, but this was not correlated with the plant phylogeny (Table 1). Denitrification ultimately causes the loss of bioavailable N, which has an adverse effect on plant growth in N-limited environments.

Although our reductive approach provided new information supporting the phylosymbiosis in plant root–microbiome associations, we recognize that the ability of the plants to control their root-associated microbial community, as described in this common garden experiment, might deviate from the natural environment. Among other reasons, the homogenized standard soil mixture used in our experiment likely comes with a distinct microbial community. However, the plant’s ability to promote a specific microbial community remains and we believe our experimental design is useful for studying the plant root–microbiome association in plants sampled globally. Indeed, at the global scale, a reciprocal transplant experiment would introduce many additional variables, such as soil, season, precipitation, and temperature variability. As noted in other studies [29,30], reductive approaches like ours pave the road for further studies investigating the effect of soil chemistry and water regime, among other factors affecting the plant–rhizosphere microbiome interaction.

Our study is also limited in that our analysis does not incorporate fungi. This reflects the paucity of reference fungal genomes in reference databases (e.g., m5nr) and the lack of bioinformatic tools to identify fungi in short-read metagenomes. However, arbuscular mycorrhizae have long been recognized as being important for plant N acquisition and other processes because they form symbiotic associations with plants. However, recent studies investigating the functional potential for N cycling in short-read metagenomes revealed that fungi accounted for few reads [4,57] or were sometimes not even detected [34]. When contributing to the N cycling potential, these fungal sequences were systematically associated with ammonia assimilation, and the assimilatory nitrate to nitrite and nitrite to ammonia pathways. In the future, the development of bioinformatic tools integrating fungal and bacterial identification tools will provide a more holistic understanding of how microbes, fungi included support N cycling in our samples and across environments.

Our reductive approach of using grasses grown in a common garden experiment removes the many potentially confounding effects of the natural environment and provides an unprecedented opportunity to connect variation in the distribution of bacterial taxa and specific functions (e.g., N cycling pathways) with the host plants’ phylogeny. Here, we report that the distribution of microbial traits involved in the production of bioavailable N is strongly affected by the plants’ evolutionary history, whereas pathways such as denitrification are not selected.

## Figures and Tables

**Figure 1 microorganisms-09-02476-f001:**
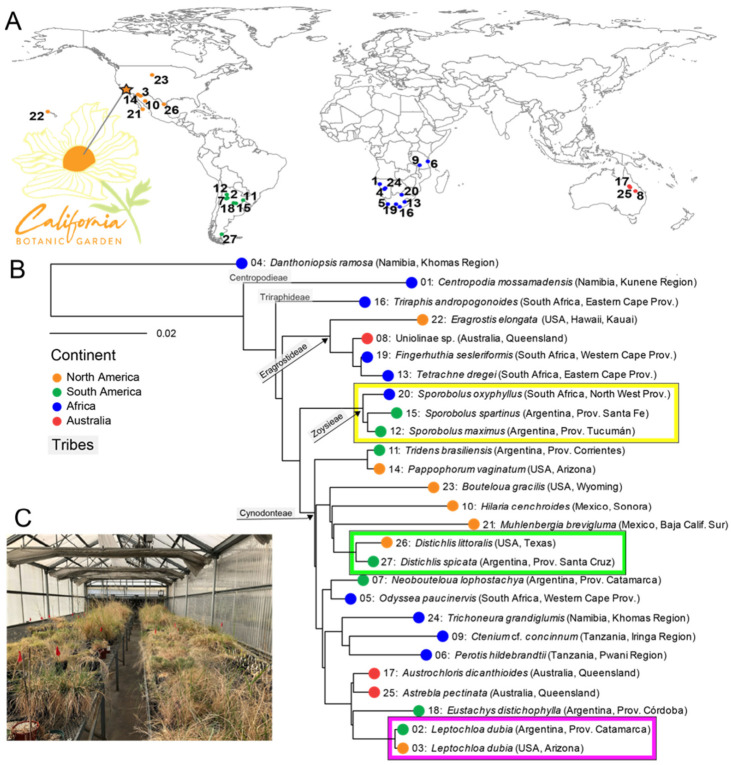
Chloridoid grass phylogeny. (**A**) Location where each chloridoid species was originally collected and the California Botanic Garden in southern California (Claremont, CA, USA), where the plants were cultivated in a greenhouse. (**B**) Maximum likelihood phylogeny of chloridoid grass species and the outgroup (*D. ramosa*) estimated using three chloroplast loci. The plants from the same species or genus (see text) are highlighted in colored boxes. The number on each branch corresponds to the collection location on the map in A. The ball on each branch is colored according to the continent of origin, so that grasses collected in North America are orange, those from South America are green, those from Africa are blue, and those from Australia are red. (**C**) Common garden experiment in the California Botanic Garden greenhouse (star in Figure 1A; see text). See Appendix A for detailed information about the plants’ origins and plant numbers.

**Figure 2 microorganisms-09-02476-f002:**
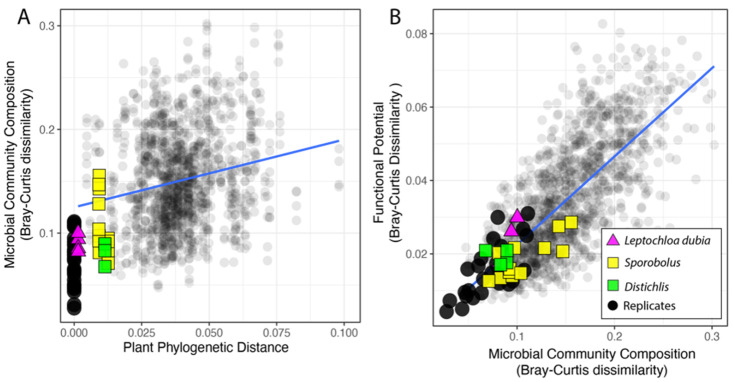
Relationship between phylogenetic distance among pairs of plants and the microbial community dissimilarity (**A**) and variation in the functional potential for N cycling in the corresponding microbial communities (**B**). Comparisons among technical replicates (same plant, black circles), samples from the same species (triangles), and from the same genus (squares) are highlighted.

**Figure 3 microorganisms-09-02476-f003:**
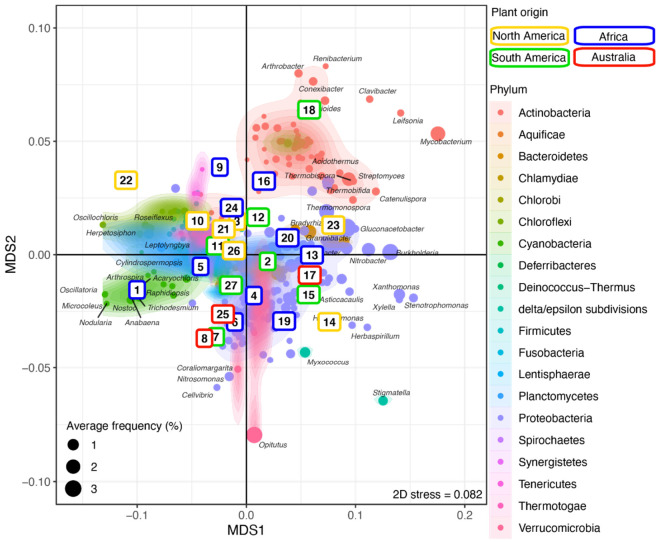
Microbial community structure across plants. NMDS ordination of the grass samples (colored by continent; see Appendix A) according to their microbial community composition. The identified bacterial genera are colored by phylum and overlaid with a 2D kernel density plot to highlight the phyla. The bacterial genera most affected by the plant phylogeny are labeled. Numbers in the colored boxes represent the plants (see Appendix A). See Appendix A for a high-resolution figure with all bacteria genera labeled.

**Figure 4 microorganisms-09-02476-f004:**
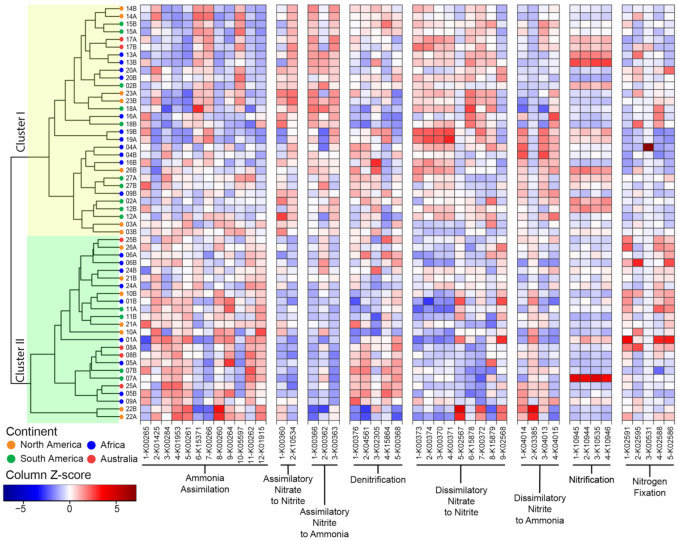
Functional potential for N cycling pathways across sequenced microbiomes. The microbiomes are identified by the plant number (see Figure 1 and Appendix A) and replicate (A or B; see Appendix A). The traits involved in N cycling are grouped by pathways and are identified by the reaction number and KO ID (see Appendix A). The Z-score normalization is by column to highlight variation in N cycling traits across samples (see Appendix A for Z-score normalization by row to highlight variation in the N cycling traits within samples). The clustering is based on the Bray–Curtis dissimilarity index and single linkages.

**Figure 5 microorganisms-09-02476-f005:**
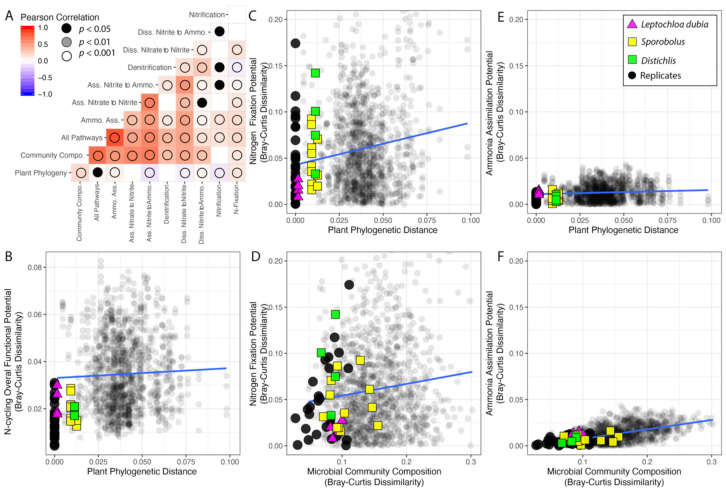
Interactions among plant phylogeny, microbial community composition, and the functional potential for N cycling pathways. (**A**) Correlogram for the N cycling pathways, rhizosphere microbial community composition, and the plant phylogeny. (**B**–**F**) Scatterplots with linear regressions showing the relationships among plant phylogeny, microbial community composition, and the functional potential for N cycling. In all of the analyses, pairwise comparison among replicate samples were systematically excluded.

**Table 1 microorganisms-09-02476-t001:** Linear regression statistics for the relationships among variation in the plant phylogeny, change in the microbial community composition (at the genus level), and variation in the distribution of traits supporting the functional potential for N cycling. Pairwise comparisons among replicate samples were systematically excluded. Significance level: *** *p* < 0.001, ** *p* < 0.01, * *p* < 0.05, NS > 0.05.

	# of Traits	Plant Phylogeny Slope (Std. Error)	Community Composition Slope (Std. Error)
Model		lm(Pathway~PlantPhylo)	lm(Pathway~Com.Comp.)
All pathways	44	0.042 * (0.021)	0.241 *** (0.004)
Ammo. Ass.	12	0.044 *** (0.012)	0.101 *** (0.003)
Ass. Nitrate to Nitrite	2	NS	NS
Ass. Nitrite to Ammonia	3	−0.197 *** (0.033)	0.335 *** (0.007)
Denitrification	5	NS	0.236 *** (0.013)
Diss. Nitrate to Nitrite	9	−0.126 *** (0.031)	0.219 *** (0.008)
Diss. Nitrite to Ammo.	4	0.141 ** (0.052)	0.249 *** (0.015)
Nitrification	4	−1.470 (0.299)	NS
N-Fixation	5	0.456 *** (0.065)	0.127 *** (0.020)

## Data Availability

Sequences generated for this project were deposited in public repositories (i.e., NCBI GenBank and MG-RAST) and are publicly accessible (see Appendix A).

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
