# Peer review of "Phylosymbiosis in the Rhizosphere Microbiome Extends to Nitrogen Cycle Functional Potential"

_microorganisms, 2021, doi:10.3390/microorganisms9122476_

Round 1

Reviewer 1 Report

It is a pleasure to read this study that is written in an excellent English, has good writing and scientific style, well planned and processed data and important conclusions.

my comments are only  minor

Abstract

line 14: this was rather a greenhouse than “common garden experiment”.

line 16: I recommend to reformulate that plant seeds were sampled making sure that this was not transplantation of whole grasses

line 21: “plant host select for specific bacterial lineages” either change to “bacterial communities” or change select to “prefer”. Because previous sentence concluded that there is conserved core of rhizosphere microbiome.

line 23: “bacterial potential for N-fixation potential” isn’t it better “bacterial ability for N-fixation potential”

Material and methods:

line 121: what is the source of bacterial innocula, peat? I think it is worth to mention of individual components of the resulting soil mixture were sterilised and if seeds were surface sterilised.

lines 128-132: I would change the order of these two sentences, first explain which loci were used for what purpose and then explain how the sequence data were gathered.

Results

line 205: “Unsurprisingly, …” please explain why unsurprisingly, because the Cynodonteae are the crown lineage?

line 206-209: “As expected” … this sentence is obvious such as next one, I would modify it that the phylogeny confirmed monophyly of members of single species or genera.

Figure 1, line 215: box for Sporobolus is missing

Figure 3, line 290: it is not explained that the numbers inside of boxes correspond to plant samples in Table S1, I would change it to “grass samples (see Table S1) colored by continent.

Discussion:

lines 415-416 sounds like duplication of line 422 concluding that “some” or “these” studies used rhizosphere. Is this here twice because the first conclusion is about rhizosphere and second about endosphere?

General discussion:

I like the text structure and information discussed.

Especially it is good that authors mentioned also different variables (line 512) and also fungal aspect (517-528).

Here are some ideas to consider:

  • it is not sure what was the vector for microbial community establishment, did also authors consider that there are some entophytic bacteria in seeds?
  • microbial communities are also formed by microbial interaction, not only plant-microbe interaction. See for example: https://doi.org/10.1371/journal.pcbi.1005366.
  • interaction between microbes and plant is based on immune suppression in roots regulated by signal proteins https://doi.org/10.1111/pce.13632. These proteins, such as others are changing under ecological and evolutionary drivers and may explain forming of different bacterial communities.
  • geological origin explain some climatic differences where the plants are adapted, but plants are also adapted to different soil conditions that may be limiting for some microbe species. This may explain why some plants are adapted and prefer some bacterial lineages
  • often it is referred to bacteria as microbiome, but there are also other organisms (fungi, etc.). At least sometimes (see e.g. line 418) authors may add “bacterial microbiome” to make sure that the discussion is about bacteria only

congratulation for the excellent research and outputs!

The Academic Editor

Author Response

Long Beach,

11/18/2021

Dear Editor,

Thank you for giving us the opportunity to resubmit this edited and improved version of our manuscript entitled "Phylosymbiosis in the Rhizosphere Microbiome Extends to Nitrogen Cycle Functional Potential" by M. Van Bel et al. that we submitted to the "Microorganisms" journal.

In this revised version of the manuscript, we have carefully considered all the suggestions made by the reviewers and incorporated many modifications in the text, the figures, and the supplementary material (see details below). All the modifications can be tracked in this new version of the manuscript.

Sincerely,

For the authors,

A. Fisher, PhD. and R. Berlemont, PhD.

The line number, when provided, in our responses (R) corresponds to the line numbers in the revised version of the manuscripts, not in the initial document.

Reviewer #1

  • It is a pleasure to read this study that is written in an excellent English, has good writing and scientific style, well planned and processed data and important conclusions.

            R: Thank you to the reviewers for their thorough comments.

  • line 14: this was rather a greenhouse than “common garden experiment”.

            R: Indeed, the experiment was conducted in a greenhouse. However, the "common garden" experiment refers to the actual type of experiment (as opposed to a reciprocal transplant experiment. In a common garden experiment, all the plants are grown under the same conditions instead of in their natural environment.

  • line 16: I recommend to reformulate that plant seeds were sampled making sure that this was not transplantation of whole grasses

            R: The text was modified to highlight that characterized microbiomes were from plants grown from the seed. (see LL16-17)

  • line 21: “plant host select for specific bacterial lineages” either change to “bacterial communities” or change select to “prefer”. Because previous sentence concluded that there is conserved core of rhizosphere microbiome.

            R: The text was modified as suggested by the reviewer. (see L22)

  • line 23: “bacterial potential for N-fixation potential” isn’t it better “bacterial ability for N-fixation potential”

            R: The sentence was modified for clarification (see L24)

  • line 121: what is the source of bacterial innocula, peat? I think it is worth to mention of individual components of the resulting soil mixture were sterilised and if seeds were surface sterilised

            R: We clarified this in the material and method section and also incorporated a section in the discussion section (see L157 and LL452-454).

  • lines 128-132: I would change the order of these two sentences, first explain which loci were used for what purpose and then explain how the sequence data were gathered.

            R: We changed the order of these sentences as the Reviewer suggested (LL 160-162)

  • line 205: “Unsurprisingly, …” please explain why unsurprisingly, because the Cynodonteae are the crown lineage?

            R: “Unsurprisingly” refers to the same issue with recovering the relationships of this clade in previous studies. We removed this since it is more appropriate for the Discussion section and the previous studies are cited there.

  • line 206-209: “As expected” … this sentence is obvious such as next one, I would modify it that the phylogeny confirmed monophyly of members of single species or genera.

            R: We revised these sentences as the Reviewer requested to make them more concise, but we wish to include the species and genera that have multiple samples to highlight the consistency of the results.

  • Figure 1, line 215: box for Sporobolus is missing

            R: This figure has been slightly modified to better highlight the boxes (see Figure 1)

  • Figure 3, line 290: it is not explained that the numbers inside of boxes correspond to plant samples in Table S1, I would change it to “grass samples (see Table S1) colored by continent.

            R: Thank you for pointing this out. The legend has been modified and now contains this important information (see L355).

  • lines 415-416 sounds like duplication of line 422 concluding that “some” or “these” studies used rhizosphere. Is this here twice because the first conclusion is about rhizosphere and second about endosphere?

            R: The Reviewer is correct. To clarify the meaning we have removed the second iteration of the sentence and included the reference in the first sentence.

Here are some ideas to consider:

  • it is not sure what was the vector for microbial community establishment, did also authors consider that there are some entophytic bacteria in seeds?

            R: We allowed the microbial community to develop from the greenhouse environment, the soil mixture consistently used, and the seeds. It is a possibility that bacteria were present in or on the seeds and they most likely affect the system, but we did not test for the presence of bacteria in the seeds. We clarified this in the discussion section (see LL452-454).

  • microbial communities are also formed by microbial interaction, not only plant-microbe interaction. See for example: https://doi.org/10.1371/journal.pcbi.1005366.

            R: This is a very interesting point and would be an interesting avenue for further study. However, it is difficult to infer specific interactions between members of a complex microbial community without assembling the MAGs and knowing the precise gene content of each member of the community.

  • interaction between microbes and plant is based on immune suppression in roots regulated by signal proteins https://doi.org/10.1111/pce.13632. These proteins, such as others are changing under ecological and evolutionary drivers and may explain forming of different bacterial communities.

            R: We recognize that this is a very interesting (and important) aspect of the plant:microbiome interaction. This specific aspect of the system is not addressed here. However, the data produced in this experiment could (should) be used to investigate how other traits could affect the interaction between plant and microbes in the phylogenetic context.

  • geological origin explain some climatic differences where the plants are adapted, but plants are also adapted to different soil conditions that may be limiting for some microbe species. This may explain why some plants are adapted and prefer some bacterial lineages

            R: Indeed, we tried to briefly investigate this aspect (e.g., figure S2), but as the reviewer mentioned, most plants have a range of preferred environmental conditions and are not just limited to a specific environment. Yet, it would be interesting to grow (some of) these plants in multiple (soil) conditions and sequence their root:microbiome to investigate the robustness of the phylogenetic signal across a range of environments and plant physiological responses.

  • often it is referred to bacteria as microbiome, but there are also other organisms (fungi, etc.). At least sometimes (see e.g. line 418) authors may add “bacterial microbiome” to make sure that the discussion is about bacteria only

            R:  We totally agree with this, and we included a specific statement about the importance and challenge of studying non-bacterial members of the soil microbiomes (see LL 615-626). In addition, as suggested by the reviewer, we used the more precise "bacterial microbiome" terminology in many instances thorough the text, from the abstract to the discussion (e.g., L19, L20, L22, L454, L468, 471, ... )!

Reviewer 2 Report

The manuscript by Van Bel et al. outlines a rigorous and substantive metagenomic analysis of the bacterial communities in the root environment of 26 choridoid grass species obtained from global locations. The manuscript is generally very well written, and the work is of interest. The specific conclusions supported by the data are, unfortunately, not clear as there are critical gaps in the methodological details, specifically pertaining to growth conditions and soil sample procurement.

The microbial community reflected in the analyses presented is the result of 26 grass species growing from seed in a soil matrix. The community dynamics leading to the community as obtained at sampling time would be driven by lant impact through diverse root exudates, as well as the pre-existing microbial community. The answers to two critical questions should then shape the interpretations of the data.

  1. Was the soil mix used (perlite, sand, finely screened peat) sterilized in advance, or was there an intentional inoculation with an existing microbiota? If the potting soil was sterile, there was no uniform microbial starting point but rather a virgin environment.
  2. Were the seeds surface sterilized? Even if they were, did they contain bacterial endophytes? Both surface-associated and seed-internal microbiota would act as inoculum to the soil. If the source/ species specific seeds were not free from bacteria, then each pot would be seeded with its own unique microbiota. This effect would be unchallenged if the potting soil was sterile.

Without this critical information I am not able to review the results and discussion sections. This is not a “dealbreaker”, just a request to revisit with more information. The authors are requested to clarify these details in a revised manuscript, and to interpret the results through the respective lens.

The authors are requested to revisit statements derived from specific gene quantities of the nitrogen cycle. Just because a gene / group of genes such as those encoding nitrogenase are present, this cannot be taken to mean they are expressed. We have a current dataset from a grassland where nif gene abundance is high, but we do not detect much nifRNA. Of course, our system is quite different on several counts, and cannot be taken to inform your system. Nevertheless, there is the precedent that cautions over-interpretationß.

Specific items:

  1. Mention of the extensive root-associated microbiome work in Arabidopsis should be included in the introduction. Clearly Arabidopsis is no grass, and this should not be viewed as decreasing the value of the work presented here. Yet the Arabidopsis work does present substantial insights into effect of the plant on shaping of the root-associated microbiome.
  2. Lines 65 – 73. Please make it clear to the reader that this section addresses microbial communities in general – at least I assume so as ref 41 on phylosymbiotic signal is based largely on human microbiome.
  3. Line 79: The selection claim is a stretch. While plants do modulate the root-associated microbiome, the pre-existing associated soil microbiome also contribute to factors impacting the community composition over time. Instead of “selection” say something like “impact”.
  4. Please provide details on how the “rhizosphere” samples were taken (section 2.2). For example, did these samples include loosely adhering soil, or were roots gently washed before root-adhering biota were removed by sonication?
  5. Line 206: What data is referred to here? I do not see any in Fig 1.
  6. Line 210: This may be a question for the editorial staff, but are logos permitted in a paper in MDPI (Fig. 1a).
  7. Line 402: Please expand on why you state the “assembly is not random”.
  8. Figures 2, 3 & 4 are either identical to, or very similar to supplemental figures S2, S3 and S4. For example Fig 4 has the identical legend and tree at left, but the colors in the boxes are different. This may simply be due to some error in which final files were uploaded.

Author Response

Long Beach,

11/18/2021

Dear Editor,

Thank you for giving us the opportunity to resubmit this edited and improved version of our manuscript entitled "Phylosymbiosis in the Rhizosphere Microbiome Extends to Nitrogen Cycle Functional Potential" by M. Van Bel et al. that we submitted to the "Microorganisms" journal.

In this revised version of the manuscript, we have carefully considered all the suggestions made by the reviewers and incorporated many modifications in the text, the figures, and the supplementary material (see details below). All the modifications can be tracked in this new version of the manuscript.

Sincerely,

For the authors,

A. Fisher, PhD. and R. Berlemont PhD.

The line number, when provided, in our responses (R) corresponds to the line numbers in the revised version of the manuscripts, not in the initial document.

Reviewer #2

The manuscript by Van Bel et al. outlines a rigorous and substantive metagenomic analysis of the bacterial communities in the root environment of 26 choridoid grass species obtained from global locations. The manuscript is generally very well written, and the work is of interest. The specific conclusions supported by the data are, unfortunately, not clear as there are critical gaps in the methodological details, specifically pertaining to growth conditions and soil sample procurement.

            R: Thank you to the Reviewer for their thoughtful and thorough comments.

The microbial community reflected in the analyses presented is the result of 26 grass species growing from seed in a soil matrix. The community dynamics leading to the community as obtained at sampling time would be driven by Plant impact through diverse root exudates, as well as the pre-existing microbial community. The answers to two critical questions should then shape the interpretations of the data.

  • Was the soil mix used (perlite, sand, finely screened peat) sterilized in advance, or was there an intentional inoculation with an existing microbiota? If the potting soil was sterile, there was no uniform microbial starting point but rather a virgin environment.

            R: As per the manuscript, the potting soil was homogenous but not sterilized and there was no intentional inoculation with existing microbiota. (see L157 and LL452-454)

  • Were the seeds surface sterilized? Even if they were, did they contain bacterial endophytes? Both surface-associated and seed-internal microbiota would act as inoculum to the soil. If the source/ species specific seeds were not free from bacteria, then each pot would be seeded with its own unique microbiota. This effect would be unchallenged if the potting soil was sterile. Without this critical information I am not able to review the results and discussion sections. This is not a “dealbreaker”, just a request to revisit with more information. The authors are requested to clarify these details in a revised manuscript, and to interpret the results through the respective lens.

            R: It is unclear if this is "critical" or "not a dealbreaker". This comment is confusing but we wish to clarify our Methods to the Reviewer and readers.  

            In brief, the soil was not sterilized and we revised the manuscript to highlight the unsterile nature of our soil and seeds (see L157 and LL 452-454)

            We disagree with the Reviewer that the initial environment needed to be sterile for several reasons. Indeed, our experimental design was aimed at reducing the complexity of the plant:microbiome system, and yet the system was still very complex! The seeds were not surface sterilized before planting. They were collected in situ, kept dry during transport, and then planted at the greenhouse. Thus, any external or internal bacteria on the seeds would have acted as the inoculum, just as in a natural setting. The phylogenetic patterns of these external and internal bacteria are what we wished to test for in this experiment. . Indeed, all these plants were grown in the same non-natural environment with the same initial soil microbiome which is most likely very different from their "natural microbiome". After rarefaction of the taxonomically identified reads, the vast majority of the identified microbial genera were found systematically across all the samples. In addition, the variation in their distribution and abundance across samples, matched significantly with the phylogenetic distance between plant host! Thus, although many factors (including some endophytes) can influence the actual composition of the analyzed microbiomes, our results can be summarized as follows (i) the plants can modulate their associated bacterial microbiome, (ii) not all the functions are "selected" similarly, and (iii) this ability is phylogenetically conserved among plants (i.e., more closely related plants tend to have more similar bacterial microbiomes).

            Finally, although this comment is clearly about a different study, it is important to note that (i) most soil sterilization approaches that do not alter the chemistry of the soil require Gamma-rays (and even then they may not be completely sterile), (ii) the idea of inoculating the soil is only "easy" in theory, and only makes sense if the soil was "sterile" (it was not). In practice, the question would be "what would be the inoculum" (composition and distribution-wise)? Maybe one could innoculate with a well-characterized "soil microbiome" but how can you "produce" this? Alternatively, "a mix of cultivable bacteria/fungi" (but again, beside the bias towards cultivable isolates that it would introduce, which microbes, what proportion), something else? Thus, we believe that the inoculum approach although very interesting for some problems related to agriculture and soil restoration is conceptually challenging for questions about phylogenetic patterns.

  • The authors are requested to revisit statements derived from specific gene quantities of the nitrogen cycle. Just because a gene / group of genes such as those encoding nitrogenase are present, this cannot be taken to mean they are expressed. We have a current dataset from a grassland where nif gene abundance is high, but we do not detect much nifRNA. Of course, our system is quite different on several counts, and cannot be taken to inform your system. Nevertheless, there is the precedent that cautions over-interpretationß.
  • Mention of the extensive root-associated microbiome work in Arabidopsis should be included in the introduction. Clearly Arabidopsis is no grass, and this should not be viewed as decreasing the value of the work presented here. Yet the Arabidopsis work does present substantial insights into effect of the plant on shaping of the root-associated microbiome.

            R: The manuscript includes references about the microbiome in Arabidopsis (e.g., Lundberg et al. Nature 2013) and we have newly included references to Duran et al 2018 and Huang et al. 2019 (see LL 76-78)

  • Lines 65 – 73. Please make it clear to the reader that this section addresses microbial communities in general – at least I assume so as ref 41 on phylosymbiotic signal is based largely on human microbiome.

            R: Yes, this section is about the concept of phylosymbiosis and is not specific to any system. The section even mentions the "mammal gut microbiome" which is not about the plant...  (see LL79-89)

  • Line 79: The selection claim is a stretch. While plants do modulate the root-associated microbiome, the pre-existing associated soil microbiome also contribute to factors impacting the community composition over time. Instead of “selection” say something like “impact”.

            R: We revised the sentence to read “While plants may impact particular microbial lineages for their rhizosphere...” (see L93)

  • Please provide details on how the “rhizosphere” samples were taken (section 2.2). For example, did these samples include loosely adhering soil, or were roots gently washed before root-adhering biota were removed by sonication?

            R: Thank you for pointing out this omission. We added a comment in section 2.2 explaining how the soil was sampled (LL167-168)

  • Line 206: What data is referred to here? I do not see any in Fig 1.

            R: We are not sure what this comment refers to.

  • Line 210: This may be a question for the editorial staff, but are logos permitted in a paper in MDPI (Fig. 1a).

            R: As per the reviewer, this aspect needs to be addressed by the editorial staff.

  • Line 402: Please expand on why you state the “assembly is not random”.

            R: The fact that closely related plants (same species, same genus) share more similar microbiomes than more distantly related plants suggests that the microbiome assembly is not random. If it was random there would be no signal, and the microbiome of closely related plants or distantly related plants would be (on average) equally distant. This is not the case here. This is explained in detail in the results section (See LL106-136).

  • Figures 2, 3 & 4 are either identical to, or very similar to supplemental figures S2, S3 and S4. For example Fig 4 has the identical legend and tree at left, but the colors in the boxes are different. This may simply be due to some error in which final files were uploaded.

            R: We believe the Reviewer may have overlooked the differences in these figures. Specific statements were added to figure S2, 3, S3, 4, and S4 to highlight the differences.

            Specifically, figure 2 is showing the relation between phylogenetic distance among pairs of plants and the microbial community dissimilarity with the plant phylogeny emphasized by colored symbols whereas S2 is showing these results with the plant geographyemphasized by colors. These figures highlight different variables potentially influencing the microbial community composition.

Next, figure 3 is showing a "simplified" version of the figure (for clarity) whereas figure S3 is a high-resolution figure with all of the genera labeled.

Finally, figure 4 is showing the clustering of the microbiome samples according to the distribution of the N-cycling traits across samples and the heatmap showing the distribution of each N-cycling trait normalized across samples whereas S4 is showing the same information but the heatmap shows normalization of the N-cycling traits within samples.

Round 2

Reviewer 2 Report

Thank you for the revised manuscript and responses to my queries.